# Ammonia Emissions from Differing Manure Storage Facilities at Two Midwestern Free-Stall Dairies

**Richard H. Grant** \*  **and Matthew T. Boehm**

Department of Agronomy, Purdue University, 915 W. State St., West Lafayette, IN 47907, USA;
mtboehm@hotmail.com
\* Correspondence: rgrant@purdue.edu

**Abstract:** Dairies contribute a major portion of agricultural ammonia emissions in the United States. Emissions were monitored over two years from an anaerobic lagoon receiving manure from cows in the milking parlor and holding area in Indiana (IN), USA and a storage basin receiving manure from cows in barns as well as the milking parlor and holding area in Wisconsin (WI), USA. Emissions were monitored using open-path tunable diode lasers, sonic anemometers, and two emission models. The mean annual daily emissions at the WI storage basins ($30 \ \mu g \ m^{-2} \ s^{-1}$) was nearly equal to that at the IN lagoon ($27 \ \mu g \ m^{-2} \ s^{-1}$). The mean annual daily ammonia ($NH_3$) emissions on a per animal basis were greater at the WI basins ($33 \ g \ NH_3 \ hd^{-1} \ d^{-1}$; $26 \ g \ NH_3 \ AU^{-1} \ d^{-1}$) (hd = animal; AU = 500 kg animal mass) than at the IN lagoon ($9 \ g \ NH_3 \ hd^{-1} \ d^{-1}$; $7 \ g \ NH_3 \ AU^{-1} \ d^{-1}$). Emissions from both storage systems were highest in the summer, lowest in the winter, and similar during the spring and fall. Emissions were strongly correlated with air temperature and weakly correlated with wind conditions. Greater emissions at the WI basins appeared to be related primarily to the characteristics of the stored manure.

**Keywords:** ammonia; dairy; manure; emissions; waste; Lagrangian Stochastic; radial plume mapping

## 1. Introduction

While nitrogen is critical to life, excess nitrogen (N) in the environment has a wide range of negative impacts including the degradation of air, soil, and water resources. Emission inventories for the United States estimate that the agricultural production sector of the economy contributes the largest portion of the total $NH_3$ emissions, with livestock production being the dominant source of $NH_3$ within this sector [1]. Ammonia emissions from livestock production result from the incomplete utilization of feedstock N in livestock growth and sustenance. Between 50% and 80% of N intake is excreted in manure (urine and feces) [2,3].

Dairy and cattle production accounts for about 40% of the national $NH_3$ emission inventory [3]. Across the United States, a wide range of production management systems are used in dairies. Housing facilities at eastern United States dairies include tie-stall, free-stall, and bedded pack barns [4]. Manure, composed of both urine and feces, and bedding is stored either combined as a slurry (urine and feces) or separated into liquid (less than 5% dry matter (DM)) and solid components. Liquid waste storage remains are considered relatively well-mixed in lagoons compared to slurry storage (7–12% DM) [4]. Ammonia emissions from the liquid storage result from rapidly hydrolyzed urea in the liquid and more slowly mineralized organic N. Emissions from the slurry storage have a significantly higher mineralization component than liquid storage due to the much greater DM. Differences in manure storage practices can be expected to influence the timing and magnitude of $NH_3$ emissions over the course of days and weeks [5].

The complexity of dairy operations and the wide range of climates in which dairies are located have resulted in a wide range of different management schemes across the US. National inventories of and regulatory constraints on dairy and cattle $NH_3$ emissions require well-documented measurements on as many management systems as possible. There have been several long-term studies of $NH_3$ emissions from manure storage systems of dairies in the arid western US [6–9], but relatively few in the eastern US [4,10]. Although emission models to evaluate different manure management systems are available [4], the validation of these models is challenging due to a limited number of long-term near-continuous $NH_3$ emission measurements. Ammonia emissions from two Wisconsin free-stall dairies with slurry storage had $NH_3$ emissions averaging 54 g $hd^{-1}$ $d^{-1}$ (hd = animal head) during the summer and 24 g $hd^{-1}$ $d^{-1}$ in the fall [10]. Slurry storage at a dairy in Canada yielded emissions ranging from 8 $\mu g$ $m^{-2}$ $s^{-1}$ (6 g $hd^{-1}$ $d^{-1}$) during the winter to 48 $\mu g$ $m^{-2}$ $s^{-1}$ (39 g $hd^{-1}$ $d^{-1}$) during the spring [5]. For regulatory and inventory purposes, it is desirable to limit the necessary classification of manure storages in $NH_3$ emission estimates. The National Air Emissions Monitoring Study (NAEMS) of emissions from dairy operations, including these two dairies, was conducted to increase the available emission measurements for such regulatory and inventory efforts [11]. It is hypothesized that although the timing of emissions varies between anaerobic lagoon and slurry storage, the seasonal and annual emissions will be similar (null hypothesis). Here, we compare the $NH_3$ emissions over a year from two free-stall dairies in similar eastern US climates using differing manure storage systems: an anaerobic lagoon storing liquid manure and pits storing slurry. Both dairies were measured as part of the NAEMS.

## 2. Experiments

Ammonia emissions from waste storage facilities on two free-stall dairies in the Midwest: two waste storage basins in Wisconsin (WI) and an anaerobic waste lagoon in Indiana (IN). Because the waste handling of each dairy operation varies, emissions were estimated in terms of the surface area of the storage facility, the number of cows producing manure for each storage (head = hd), and the number of animal units (1 AU = 500 kg) producing manure for each storage area.

The WI free-stall dairy consisted of five barns for lactating Holstein cows, a special needs barn, a feed storage area, and a milking parlor [12,13]. Ammonia emissions were measured from the northernmost and middle clay-lined storage basins (combined volume of 16,980 $m^3$ and surface area of 7091 $m^2$) that received wastewater from the flush of two of the barns, the holding area, and the milking parlor. The northernmost basin received liquid waste from two barns after solids separation when the separator was functioning and received both solids and liquids when the separator failed (Figure 1a). The middle basin received flushed waste from the holding area and milking parlor without solids separation. The third (southernmost) basin filled when the other two basins exceeded capacity and rarely had any manure in it. The 3.3 m deep basin had a sludge depth estimated at 0.3 m. Liquid manure was removed from the basins approximately every 12 weeks with the solids removed twice a year. Assuming manure deposition was uniform throughout the day, a 24 h day time-weighted cow population loading the basins consisted of the time for all lactating cows (1506 hd) to be milked (three times a day for 30 min each) and the time that cows in two barns (496 hd) were present in those barns. The time-weighted population loaded the basins was 559 lactating cows [13]. Given the average mass of a cow, the equivalent animal unit was 710 AU.

The IN free-stall dairy consisted of three barns for lactating Holstein cows, a feed storage area, a special needs barn, and a milking parlor [13,14]. The 4.8 m deep clay-lined waste lagoon had a maximum volume of 48,212 $m^3$ and a surface area of 9884 $m^2$ (Figure 1b). The anaerobic lagoon received wastewater (flush) from the holding area and milking parlor. All solid waste from the holding area and milking parlor was transferred into a settling pit draining into the anaerobic lagoon (a weir limiting the solids transferred into the lagoon). The solids from this pit were removed every four to five weeks. All liquid waste was transferred by pipes to the northeast corner of the lagoon. Liquid was removed from the lagoon every 7 to 14 days except when the fields were frozen. The dairy was completed in 2002 and had not had any sludge removed from the lagoon through the time of

measurements. The facility had an average population of 2580 lactating cows (3509 AU) over the study period [14]. The mean cow mass was 636 kg.

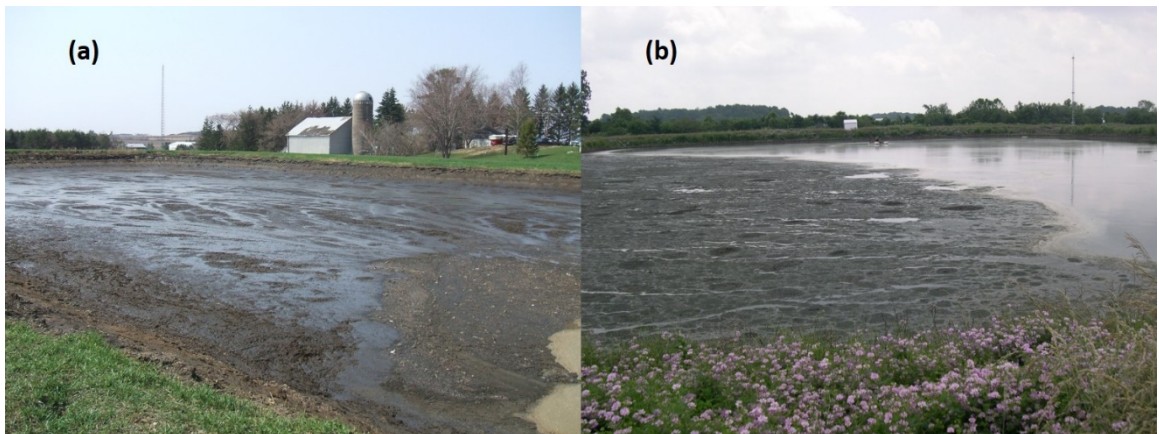

**Figure 1.** Manure storage facilities. The northern Wisconsin (WI) basin in spring 2008 with a visually estimated 85% crust on the surface (**a**) and the Indiana (IN) anaerobic lagoon in spring 2009 with visually estimated 60% scum and foam on the surface (**b**).

Meteorological measurements of barometric pressure (278, Setra, Inc., Boxborough, MA, USA), air temperature and relative humidity (HMP45C, Vaisala, Inc., Helsinki, Finland), and solar radiation (210, LiCOR, Inc., Lincoln, NE, USA) were made on a mast located on the berm 5 m west and 30 m north from the southwest berm corner at the IN dairy and between the northernmost and middle basin in the berm center 20 m east of the northwest corner of the middle basin at the WI dairy. The lagoon conditions were characterized by measurements of pH (CSIM11, Innovative Sensors, Inc, Campbell Scientific, Logan, UT, USA.), oxidation reduction potential (CSIM11-ORP, Innovative Sensors, Inc.), and temperature (107-L, Campbell Scientific, Logan, UT, USA) at a 0.3 m depth. Meteorological and lagoon characterization measurements were collected by a data logger (Model CR1000, Campbell Scientific, Logan, UT, USA) and telemetered to the on-site instrumentation trailer. Lagoon characterization sensors were removed from the lagoons when freezing conditions were expected. Since the temperature of the lagoon and basins was not measured throughout the study period, air temperature was used as a surrogate. All gas concentrations were corrected to 20 °C and a barometric pressure of 101.3 kPa.

Manure storage systems were visually assessed for surface condition (including if frozen and what fraction of surface was covered with scum, foam, or crust) by walking around the storage during each visit by the field personnel (Table S1; Figure 1). The measurement of emissions required the measurement of gas concentration and wind. The path-integrated concentration (PIC) of $NH_3$ was measured using tunable diode laser spectrometers (TDLAS; GasFinder2®, Boreal Laser, Ltd., Edmonton, AB, Canada). Two open-path, monostatic TDLAS instruments were mounted on opposite corners of each monitored lagoon or basin [12,14]. Each instrument scanned five retroreflectors on each of two adjacent sides of the lagoon or basin, three optical paths (OPs) along the berm at 1 m above ground level (agl), and two OPs defined by the scanner at 1 m agl and retroreflectors on towers at 7 m and 15 m agl. Concentrations were calculated for a given path $i$ as $PIC_i/L_i$ where L is the length of OP. The mean wind speeds and turbulence statistics were obtained from 3D sonic anemometers (Model 81000, RM Young Inc., Traverse City, MI, USA) mounted at 2.5 m agl on the meteorological mast and recorded at 5 min intervals. To be considered a valid 5 min period, at least 90% (4320) of the possible 4800 16 Hz values had to be present and the sonic temperature variance (a general measure of sensor performance) had to be less than 2.5 $K^2$. The valid 5 min statistics were then averaged to ½ h intervals. No coordinate rotations were made on the sonic anemometer measurements as the flow was likely influenced by the height changes in the crop and lagoon liquid surface at the IN dairy and height changes with wind direction due to buildings, trees, a downward slope to the west of the berm, and

varying heights of the liquid surface in the basins at the WI dairy. Variations in surface roughness with wind direction were evaluated using the turbulence characteristics derived from the sonic anemometer.

Emissions were estimated using two models: a backward Lagrangian Stochastic (bLS) model (WindTrax; Thunder Beach Scientific, Nanaimo, BC, Canada, http://www.thunderbeachscientific.com) and a vertical radial plume measurement (VRPM) emission model. The bLS model [15] quantifies the ratio between the difference between the downwind and upwind (background) concentrations $(C_i - C_{BG})$ and the average surface flux density across the source area $(F_{c,0})$ with the assumption that the ratio $(C_i - C_{BG})/F_{c,0}$ is only a function of flow characteristics. To determine the relationship between $C$ and $F_{c,0}$, the model simulates the flight path of air parcels backwards from the line sample until each parcel intersects the ground (a "touchdown"). Our application of the bLS model had more concentration measurements than source emissions to be estimated, so the emission rate $(F)$ of the one source and $C_{BG}$ for the interval of measurement was derived from a set of concentration measurements by simultaneous solution of equations for each line measurement using a standard Singular Value Decomposition algorithm. Emissions were excluded from analysis if: (1) $u_*$ was less than 0.15 m s$^{-1}$, (2) the Monin–Obukhov length |L| was less than 2 m, (3) the fraction of touchdowns within the source area was less than 0.2, or (4) $C_{BG}$ was greater than 0.15 μmol mol$^{-1}$. The bLS model's emission minimum detection limit was estimated from the calculated emissions when each storage facility was frozen. The bLS model had a theoretical random error of 22% [16] and a tracer-estimated error between 5% and 36% [17]. A comparison between the bLS model and an integrated horizontal flux model at these locations resulted in an emission underestimation of 6% by the bLS model [18]. The bLS error assumed for this study was 20%.

The VRPM emission model, an integrated horizontal flux method, calculates the emission from a defined source utilizing a measured wind speed profile and measured path-integrated gas concentrations [18]. The method utilized all five OPs along each side of the manure storage area and assumed a bivariate Gaussian function to describe the distribution of mass across the vertical plane [19]. Emission estimates were excluded using a number of quality assurance indicators designed to assure the measurement plane contained at least 70% of the plume [19]. These indicators included: (1) a Concordance Correlation Factor (CCF) less than 0.8, (2) a plume position correction factor (A) less than 0.9, and (3) a topmost PIC more than 90% of the midlevel PIC. CCF is equal to rA where A is a correction factor for the shift in the plume mass location relative to the measured PIC and equal to CCF/r (r is the Pearson correlation coefficient between CCF and A). Prior studies evaluating the accuracy of the VRPM method using five OPs and relatively stringent quality assurance criteria ranged from underestimates of 21% +/− 14% [20], 6% [19], and 19% +/− 33% [21] to overestimates of 10% +/− 16% [22] and 46% +/− 26% [23]. Grant et al. concluded that the mismatch of a relatively slow response NH$_3$-measuring TDLAS and faster response sonic anemometers resulted in the overestimation of emissions [18] comparable to the 5% to 20% estimated by Denmead [24].

The bLS and VRPM emission estimates were considered to be independent measurements of the emissions during any given ½ h averaging period. Since the estimated errors for the bLS emission model and the VRPM emission model were comparable at approximately 20%, emissions determined using the two methods were considered comparable but differing in a specific situation due to the characteristics of local turbulence [18]. A site-specific comparison of the emissions estimated using the two methods provided a means of estimating the influence of site-specific turbulence [18] (Table S2). Although it is unknown whether the bLS or VRPM emission model is more accurate, we adjusted the bLS emission estimates to VRPM-equivalent emissions according to the intercomparison of the two methods at the two dairies (Table S2). All bLS emission estimates were combined with the VRPM emission estimates (if present) to determine the best estimate of the daily and seasonal emissions.

A fetch-to-height ratio criterion of 100:1 defined valid upwind turbulent conditions for emissions calculation. There were no upwind obstructions at the IN dairy lagoon that limited the calculation of emissions based on wind direction [13]. The farm buildings and trees to the north and east of the basins at the WI basin limited the valid emissions to wind directions from 180° to 360° [12]. After the

exclusion of wind directions previously stated, the surface roughness (represented by the calculated $z_o$) at the IN basins and WI lagoon was nearly identical (0.12 m ± 0.167 m (mean ± SD) at WI and 0.13 m ± 0.177 m at IN).

The NH$_3$ emissions were assumed to be stationary and resulting primarily from dissolved NH$_3$ in relatively dilute water [25]. The combined temperature influence on daily emissions associated with the biological production of NH$_4{}^+$, the dissociation of NH$_4{}^+$ to NH$_3$, and the solubility of NH$_3$ concentrations were evaluated for all days on which the storage surface was not frozen using an exponential van't Hoff function often used to describe the temperature influence on the air: the liquid diffusion process of a gas:

$$F_{NH3}(T_{air}) \;=\; \alpha \, e^{-\beta \left( \frac{1}{T_{air}} - \frac{1}{298.15} \right)} \tag{1}$$

where $\alpha$ represents the source strength (units of mass per m$^2$, hd, or AU per unit time), $\beta$ represents the temperature influence on the emission (set to 4100 [26]), and the mean air temperature ($T_{air}$) is in K. Since lagoon temperature was measured for only a relatively small portion of the measurement, air temperature was used as a proxy of the air–liquid interface temperature. The use of air temperature as a proxy for near-surface liquid temperatures was used in describing NH$_3$ emissions for hog operations [27].

Comparisons between seasons at a given dairy or between storages at the two dairies require consideration of the environmental differences between the dairies. The temperature influence on emissions was minimized by normalizing the measured emissions ($Q_{NH3}$) to those predicted at 25 °C (Equation (1)) according to:

$$Q_{NH3}(25,U) = Q_{NH3}(T_{air},U) \, F_{NH3}(25\ °C)/F_{NH3}(T_{air}) \tag{2}$$

where $U$ is the mean measured wind speed. Transport of the gas into the air over the storage facility requires turbulent or molecular mixing of the air directly at the surface to the air overlying the storage facility.

To minimize the influence of differences in site-specific turbulent mixing on the measured NH$_3$ emissions, the temperature-normalized NH$_3$ emissions $Q_{NH3}(25,U)$ (Equation (2)) were linearly regressed on $U$ as a measure of the atmospheric mixing. A normalized emission minimizing the influence of both temperature ($T_{air}$ = 25 °C) and wind conditions ($U$ = 2 ms$^{-1}$) was defined as:

$$Q_{NH3}(25,2) = Q_{NH3}(25,U) \, [\gamma(2) + \eta]/[\gamma \, (U) + \eta] \tag{3}$$

where coefficients $\gamma$ and $\eta$ are determined by linear regression of $U$ on $Q_{NH3}(25,U)$ [27].

The relative emission error was estimated assuming independence of the errors associated with the bLS emission model, the measurement error of each gas analyzer, and, where appropriate, the error associated with estimating the delta concentration across the lagoon/basins. Estimates of daily emissions were based on days for which at least 50% of the hourly measurements were valid within a 24 h day [26]. Emission errors were estimated by dividing the relative error by the square root of the minimum number of hourly values needed to estimate a daily emission rate. The minimum of 50% of possible half-hourly emission estimates provided an estimated daily emission error of less than 20%. Mean seasonal emissions were estimated from the mean daily emissions during the season. Annual emission estimates were determined by summing the mean seasonal emissions. The representativeness of the seasonal mean daily emissions was assessed by comparing the mean annual emission to the mean daily emissions regardless of season (statistically the total mean).

A number of manure handling and bedding changes, reported by the producers, occurred at the two dairies over the study period (Tables S1 and S2). Since these activities were discrete events, the impacts of these changes on NH$_3$ emissions were explored. Since the various events occurred during different times of the year, the potential impacts of the events were evaluated relative to changes in $Q_{NH3}(25,2)$ (Equation (3)). It was assumed that the surface of the manure storage was frozen

throughout the time period between two site visits if it was observed to be frozen each visit (Tables S1 and S2). Likewise, it was assumed that the storage was crusted throughout the time period between two site visits if it was observed to be crusted each visit (Table S1). Assessments of the association of these changes in operation and manure storage surface conditions were made using an unpaired Student's *t*-test with $\alpha = 0.05$.

The fraction of N intake lost as $NH_3$ from the manure storage facilities was assessed. The IN producer supplied the herd composition of dry, fresh, early, and late lactating herd fractions. The WI producer supplied a herd composition of dry and lactating cows. It was assumed that the lactating cows were evenly distributed between fresh (first two to four weeks after calving), early (four to fourteen weeks after calving to peak lactation), and late (after peak lactation) lactating cows. Actual feed rations were not reported by the producer. Nutritional requirements of the herds at the WI and IN dairy were estimated assuming a nominal feed ration and consumption resulting in a feed dry matter intake (DMI) of 0.05 kg kg$^{-1}$ d$^{-1}$ for fresh cows, 0.04 kg kg$^{-1}$ d$^{-1}$ for all other lactating cows, and 0.02 kg kg$^{-1}$ d$^{-1}$ for dry cows and feed crude protein (CP) of 9.9% (nonlactating cows), 12.4% (fresh), 16.7% (early), and 14.1% (late) cows [28]. The N content of CP was assumed to be 16%. Since the diet was assumed, an error bound in the percentage of N intake in the measured $NH_3$ emissions was approximated by increasing the assumed DMI by 0.01 kg/kg for early cows (within the normal range of possible diets of the high-yielding cows).

## 3. Results and Discussion

Half-hourly emission measurements were made nearly continuously from fall 2008 through summer 2009 at the IN dairy (Table S3), and over a series of two- to three-week periods from summer 2007 through to spring 2009 at the WI dairy (Table S4), resulting in a substantial overlap in the measurement periods. Over the course of the measurement periods (Table 1), the mean daily air temperature at the IN dairy varied from −14.2 °C to 27.1 °C, while that at the WI dairy varied from −21.2 °C to 22.7 °C. The temperatures at both dairies were generally within the normal climatological range for their location [12]. While the mean wind speed at each dairy was the same (2.6 m s$^{-1}$), wind speeds had a wider range at the IN dairy (0 m s$^{-1}$ to 12.7 m s$^{-1}$) than the WI dairy (0.2 m s$^{-1}$ to 7.8 m s$^{-1}$) (Table 1).

**Table 1.** Seasonal weather conditions during the 2007–2009 study.

| | Indiana | | | | Wisconsin | | | |
|---|---|---|---|---|---|---|---|---|
| | Air Temperature (°C) | | | Wind Speed (ms$^{-1}$) | Air Temperature (°C) | | | Wind Speed (ms$^{-1}$) |
| | Maximum | Minimum | Mean | Mean | Maximum | Minimum | Mean | Mean |
| Spring | 27.9 | −6.1 | 9.7 | 3.1 | 23.9 | −21.8 | 4.0 | 3.2 |
| Summer | 33.0 | 7.8 | 20.8 | 1.7 | 33.4 | 7.4 | 20.6 | 2.1 |
| Fall | 29.9 | −9.0 | 10.2 | 2.3 | 22.5 | −15.1 | 3.2 | 3.0 |
| Winter | 17.3 | −29.2 | −4.6 | 3.3 | 5.8 | −26.8 | −11.4 | 2.8 |
| Annual | 33.0 | −29.2 | 8.9 | 2.6 | 33.4 | −26.8 | 7.9 | 2.7 |

The appearance of the lagoon or basins was recorded on almost every site visit. The IN lagoon generally had less than 10% crust on the surface, while the WI basins typically had crusting or scum on the entire surface (Table S1). During the winter, the lagoon and basins were generally frozen and snow-covered (Table S1). The air temperature at the WI dairy averaged −16 °C (+/− 6 °C) during periods with an observed frozen surface. The air temperature at the IN dairy averaged −3 °C (+/− 7 °C) during periods with an observed frozen surface.

Measurements were made on a total of 337 days at the IN dairy (14,525 half-hour measurement periods) distributed fairly evenly across seasons (Table 2). Twenty percent of the measurements were excluded from the bLS estimation due to failing the atmospheric turbulence and stability criteria. All excluded VRPM measurements were due to at least one missing downwind measurement. Quality assurance criteria reduced the number of half-hourly emission estimates by 18% with 8150 bLS

estimates of emissions and 3696 VRPM estimates of emissions with the fewest valid emission estimates during the winter (Table 2). Measurements were made on a total of 176 days at the WI dairy (7855 half-hour measurement periods) distributed fairly evenly across seasons (Table 2). Twenty percent of the measurements were excluded from the bLS estimation due to failing the atmospheric turbulence and stability criteria and 27% due to excluded wind directions. Sixty-one percent of the excluded VRPM measurements were due to at least one missing downwind measurement and eight percent due to excluded wind directions (many of the periods with missing downwind measurement periods also were when the winds were from an excluded direction). Quality assurance reduced the number of half-hourly emission estimates by 62% with 2644 bLS estimates and 357 VRPM measurements. There were a total of 3001 valid half-hourly emission measurements including both bLS and VRPM estimates with half as many valid emission estimates during the winter than during the other three seasons (Table 2).

**Table 2.** Emission measurements during the 2007–2009 study.

|  | Indiana | | | Wisconsin | | |
|---|---|---|---|---|---|---|
|  | Measurements | | | Measurements | | |
|  | 1/2 h Field | Valid 1/2 h Emissions | Valid Day Emissions | 1/2 h Field | Valid 1/2 h Emissions | Valid Day Emissions |
| Spring | 3712 | 3668 | 77 | 1971 | 783 | 12 |
| Summer | 3690 | 2991 | 59 | 2557 | 837 | 7 |
| Fall | 3217 | 2820 | 58 | 1689 | 936 | 15 |
| Winter | 3906 | 2367 | 52 | 1638 | 445 | 3 |
| Annual | 14,525 | 11,846 | 246 | 7855 | 3001 | 37 |

### 3.1. Daily Emissions

A total of 37 daily emission estimates were determined for the WI basins (Table 2), including 36 bLS estimates and 1 VRPM estimate. A total of 246 daily emission estimates were determined at the WI lagoon (Table 2): 174 bLS estimates and 72 daily VRPM estimates. Most WI basin emission estimates were for emissions during the spring and fall (Table 2). The IN lagoon emission estimates were nearly evenly distributed throughout the year (Table 2).

The range of measured daily minimum 1/2 h $NH_3$ concentration ($C_{min}$) and the mean daily calculated $C_{BG}$ at the WI basins were much greater than that at the IN lagoon (Figure 2). Greater variability in the WI $C_{BG}$ and $C_{min}$ (Figure 2) was likely due to the closer proximity of the WI dairy barns to the measured storage facilities (closest barn 14 m to the east at the WI dairy and 245 m to the N at the IN dairy) resulting in the greater influence of the barns on the $C_{min}$ and $C_{BG}$ at the WI basins regardless of wind direction. The median $C_{min}$ when the surface was not frozen was 33.9 µmol mol$^{-1}$ (26 µg m$^{-3}$) at the IN lagoon and 16.1 µmol mol$^{-1}$ (12 µg m$^{-3}$) at the WI basins. The median $C_{min}$ when the surface was frozen was 11.8 µmol mol$^{-1}$ (9 µg m$^{-3}$) at the IN lagoon and 37.7 µmol mol$^{-1}$ (29 µg m$^{-3}$) at the WI basins with the relatively high median $C_{min}$ at the WI basins was likely due to the proximity of the barns.

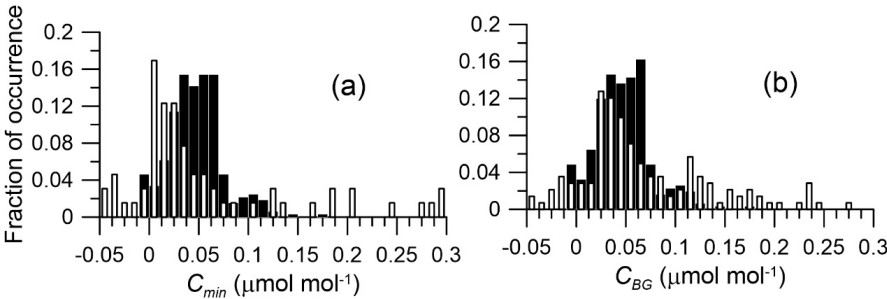

**Figure 2.** Minimum measured ($C_{min}$) and background $NH_3$ concentration ($C_{BG}$). The distribution of daily $C_{min}$ (**a**) and daily mean $C_{BG}$ derived from the backward Lagrangian Stochastic (bLS) emission model (**b**) is indicated for the IN (black) and WI (white) dairies.

Estimates of the minimum detection limit (MDL) of the bLS and VRPM emission methods were based on median concentration values rather than mean concentrations to minimize the influence of the non-normal concentration distributions (Figure 2). The MDL for the bLS emission method was estimated based on calculated emissions when the storage surfaces were frozen (Table S1). The median daily mean $C_{BG}$ (bLS-derived) was slightly higher at IN (0.047 µmol mol$^{-1}$; 36 µg m$^{-3}$) than at WI (0.040 µmol mol$^{-1}$; 30 µg m$^{-3}$). The corresponding median daily NH$_3$ emission determined by the bLS model was 10.4 µg m$^{-2}$ s$^{-1}$ for the WI basins and 15.3 µg m$^{-2}$ s$^{-1}$ for the IN lagoon. The MDL for the VRPM method was estimated assuming a constant profile of the $C_{min}$ (which would occur when there was no emission from the surface) when the surface was frozen. This corresponded to a VRPM MDL of 10.3 µg m$^{-2}$ s$^{-1}$ for the WI basins and 15.6 µg m$^{-2}$ s$^{-1}$ for the IN lagoon. The mean MDL for the combined VPM- and bLS-determined daily emissions was 10 and 13 µg m$^{-2}$ s$^{-1}$ for the WI basins and the IN lagoon, respectively.

Daily NH$_3$ emissions in terms of unit exchange surface area at the two dairies were similar (Figure 3b). Emissions during the summer may have been higher at the WI basins than the IN lagoon, although this is inconclusive due to the limited number of valid measurement days at the WI dairy. Daily mean emissions were positively correlated with $C_{BG}$ at the WI manure storage facility (linear regression R$^2$ of 0.21)(Figure 3a). Daily mean emissions were not correlated with $C_{BG}$ at the IN manure storage facility (linear regression R$^2$ < 0.01). The relationships between $C_{BG}$ and emissions at the two manure storage facilities imply that the emissions are not an artifact of the calculation method (high $C_{BG}$ corresponding to low emissions) but are a result of changing source conditions. Since the herd size and hence manure production and manure storage inputs are relatively constant from day to day, the wide range of emissions for a given period of time indicates that the environmental conditions strongly influenced the mean daily emissions.

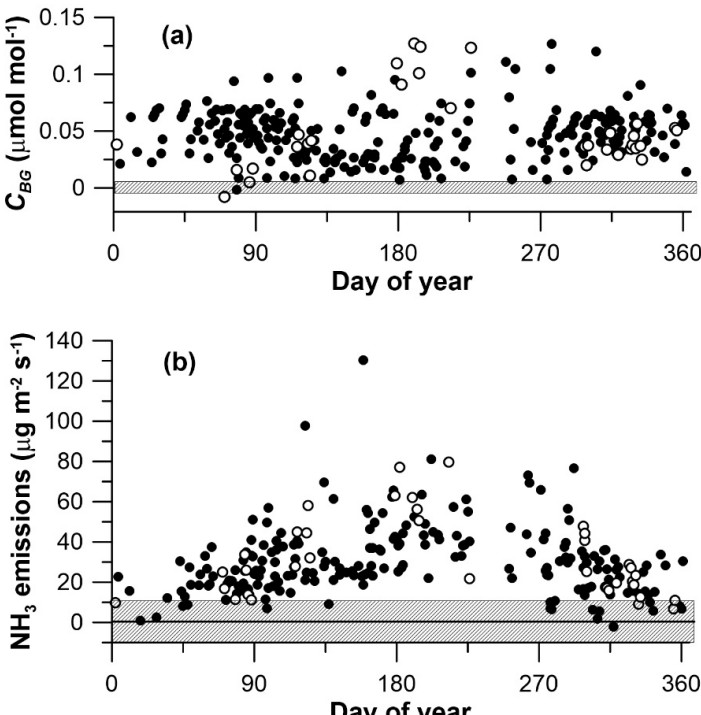

**Figure 3.** Annual variation in mean daily emissions and mean $C_{BG}$. Daily mean $C_{BG}$ associated with emissions (**a**) and mean emissions (**b**) and for the IN dairy lagoon (filled circles) and WI basins (open circles). Hatched regions represent the +/− median $C_{BG}$ during frozen conditions (**a**) and emission minimum detection limit (MDL) (**b**).

### 3.1.1. Temperature Influence

Air temperature, used as a proxy for the surface manure and lagoon temperature, was clearly related to the daily NH$_3$ emissions at both the WI and IN dairy manure storage facility (Figure 4). The exponential diffusion function of temperature (Equation (1)) accounted for 58% (adjusted R$^2$ = 0.58) of the variability in NH$_3$ emissions at the WI basins and 41% (adjusted R$^2$ = 0.41) of the variability in NH$_3$ emissions at the IN lagoon. The emission response to temperature at the IN lagoon agreed with most prior studies [5,29]. Based on the temperature response, the surface crusting of the WI nonfrozen slurry storage did not appear to restrict emissions in a similar manner as that found for a heavily crust untreated raw manure storage [5]. The fraction of explained variability in emissions at these two dairies was similar to that reported for anaerobic lagoons at OK hog operations [27].

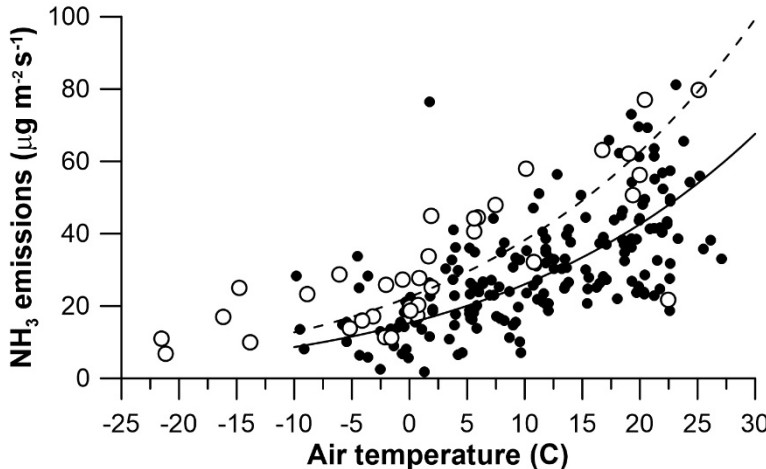

**Figure 4.** Influence of air temperature on daily NH$_3$ emissions. The influence of air temperature on daily emissions at the IN (filled circles) and WI (open circles) is indicated. The best regression fit of the form of Equation (1) for the IN dairy (solid line) and WI dairy (dashed line) for a given temperature. Three outliers were excluded from IN lagoon emission analysis: one negative and two outlying high-emission measurements (Figure 3).

The source strength ($\alpha$) in Equation (1) represents the ability of NH$_3$ at the storage surface to diffuse into the air. The source strength was significantly greater (95% confidence limit) for the WI storage (79.2 µg m$^{-2}$ s$^{-1}$, standard error 4.7 µg m$^{-2}$ s$^{-1}$) than for the IN storage (53.9 µg m$^{-2}$ s$^{-1}$, standard error 1.5 µg m$^{-2}$ s$^{-1}$). Remembering that the WI dairy manure storage included both stall and milking area manure, while the IN dairy manure storage included only waste from the milking area, the emissions need to be referenced to the manure production of the lactating herd. The source strength ($\alpha$) in terms of AU was 13.1 g AU$^{-1}$ d$^{-1}$ for the IN storage and 68.3 g AU$^{-1}$ d$^{-1}$ for the WI storage. The higher $\alpha$ at the WI basins compared to the IN lagoon indicated higher loading at the WI basins. This will be discussed later.

Diffusion across the lagoon or basin exchange surface depends on both the NH$_3$ concentration gradient and the resistance to transport. Since $C_{min}$ and $C_{BG}$ at the two storage facilities were mostly similar (Figure 2), the unmeasured NH$_3$ concentration gradient from surface to air was likely higher at the WI basins holding slurry than the IN lagoon holding largely liquid manure [30]. Manure surface NH$_3$ depends on the available NH$_4^+$, the solution temperature, and pH. While the NH$_4^+$ concentration was not measured, pH and temperature were measured (0.3 m depth). Based on a compilation of studies, it was likely that the NH$_4^+$ of the WI storage pit slurry was much higher than that at the IN lagoon; the compilation indicated a mean NH$_4^+$ -N of 1.12 kg m$^{-3}$ for slurry, 0.75 kg m$^{-3}$ for milking center manure and wastewater, and 0.38 kg m$^{-3}$ for lagoon surface water [30]. The equilibrium between NH$_4^+$ and NH$_3$ at a given temperature is evident by the solution's pH. Although only measured for limited periods of time, the differences in pH at 0.3 m in the WI basins and IN lagoon (summer pH

averaged 7.2 in the IN lagoon and 6.6 in the WI basin while fall pH averaged 8.3 in the IN lagoon and 7.1 n the WI basin) would only explain a change in the $NH_4^+$-$NH_3$ equilibrium of less than 2%. Therefore, the differences in the surface $NH_3$ between storage facilities were likely primarily a result of the differences in the characteristics of the manure held in the storage facility. The high $C_{BG}$ at the WI basins compared to the IN lagoon and the positive correlation between $C_{BG}$ and emission (previously discussed) would support the hypothesis of higher surface $NH_3$ at the WI basins compared to the IN lagoon. As previously stated, the mixing of the solution in the storage to the surface of the storage (and thus maintaining high $NH_4^+$ and $NH_3$ concentrations at the surface) would be expected to be more efficient for the liquid solution of the IN lagoon than the slurry found in the WI basins [4]. Consequently, the greater source strength of the WI basins over the IN lagoon did not appear to be associated with mixing efficiency.

### 3.1.2. Wind Influence

Gas transport into the atmosphere across the exchange surface is controlled partially by the diffusion rate through a laminar layer of air near the surface and partially by the diffusion rate through a turbulent layer of air above the laminar layer. In general, wind shear at the surface promotes the emission of $NH_3$ from the storage surface through turbulent diffusion of the gases released from solution at the liquid surface [31] or diffusion of the gas through crust or pore spaces.

A very weak linear relationship between U and $Q_{NH3}(25, U)$ occurred at the IN lagoon ($R^2 = 0.002$; Figure 5). A weak linear relationship between U and $Q_{NH3}(25, U)$ occurred at the WI basins ($R^2 = 0.06$; Figure 5). The higher correlation of $U$ to the $Q_{NH3}(25, U)$ at the WI basins than the IN lagoon suggests that the greater source strength of the WI basins may partially be a result of greater mixing in the air over the WI basins than the IN lagoon.

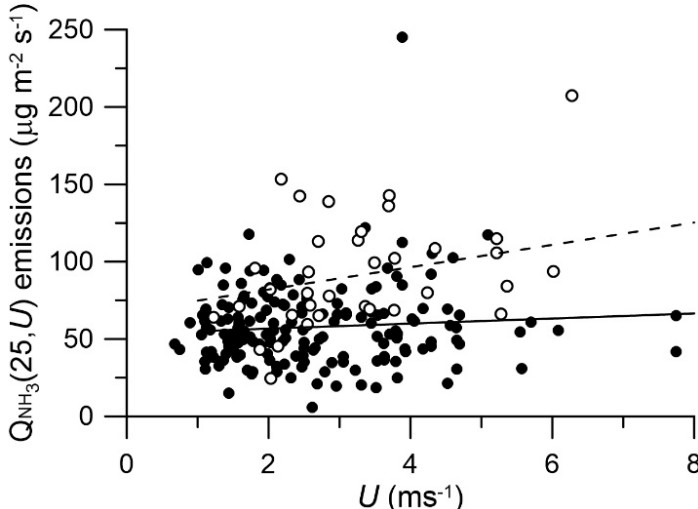

**Figure 5.** Influence of wind speed on temperature-normalized emissions. Daily mean wind speed related to the temperature-normalized (to 25 °C) daily emissions are indicated for the WI dairy basins (open circles) and IN dairy lagoon (closed circles). Linear regression fits for the WI (dashed line) and IN (solid line) are indicated. One outlier high emission value for the IN lagoon was excluded from the figure but not excluded from the analysis.

### 3.1.3. Influence of Operational Activities

The settling pit at the north end of the IN dairy lagoon was cleaned on a regular basis with nine cleaning events during measurements. Of the nine cleaning events during measurements, only six occurred under above freezing conditions and only three of these had emission estimates for all three days. Since the settling pit was outside the lagoon emission measurement domain, the cleaning events should have a negligible effect on lagoon emissions. The mean $Q_{NH3}(25,2)$ daily

emissions the day before, day of, and day after cleaning events were 55 $\mu g\ m^{-2}\ s^{-1} \pm 13\ \mu g\ m^{-2}\ s^{-1}$, 59 $\mu g\ m^{-2}\ s^{-1} \pm 13\ \mu g\ m^{-2}\ s^{-1}$, and 56 $\mu g\ m^{-2}\ s^{-1} \pm 4\ \mu g\ m^{-2}\ s^{-1}$, respectively. As might be expected due to the low sample size, there was no significant difference in emissions between the day before and the day of cleaning ($n = 5$ unpaired t of $-0.09$, critical t for two-tailed $\alpha = 0.05$ of 2.57) or between the day of and the day after cleaning ($n = 4$, unpaired t of $-0.15$, critical t for two-tailed $\alpha = 0.05$ of 2.78). The lack of influence of manure removal on $NH_3$ emissions was consistent with prior studies [32].

The liquid/solid separator for the WI dairy was also outside the basin emission measurement domain but influenced the composition of the material in the basins. The separator removed bedding and solids that had been washed from the milking area, holding area, and barns. As previously stated, the equipment frequently failed, resulting in only 12 days of valid measurements when the separator was operating and 25 days when it was not operating. Since the separator commonly failed in the winter when the surface was frozen (Tables S2 and S4), there were only 19 valid measurement days when it was not operating. The mean $Q_{NH3}(25,2)$ daily emissions with solids separation (80 $\mu g\ m^{-2}\ s^{-1} \pm 27\ \mu g\ m^{-2}\ s^{-1}$) was not significantly different (unpaired t of 0.11, critical t for two-tailed $\alpha = 0.05$ of 2.05) from that when there was no solids separation (83 $\mu g\ m^{-2}\ s^{-1} \pm 32\ \mu g\ m^{-2}\ s^{-1}$). Crusting, commonly resulting from the lack of solids separation, has been shown to affect $NH_3$ emissions (but not with statistical significance) [5]. At this dairy, the separation was never sufficient to result in no crust on the storage surface. Consequently, it is not surprising that separation had no statistically different $NH_3$ emissions.

The cow bedding material changed a few times at the WI dairy and included shavings, sand, recycled sand, and a mix of sand and recycled sand (Table S4). These changes could potentially change the emissions of the basins. The mean $Q_{NH3}(25,2)$ daily emissions were not significantly different (unpaired t of 0.11, critical t for two-tailed $\alpha = 0.05$ of 2.05) between when the bedding comprised wood shavings (12 days, 80 $\mu g\ m^{-2}\ s^{-1} \pm 27\ \mu g\ m^{-2}\ s^{-1}$) or sand (19 days, 83 $\mu g\ m^{-2}\ s^{-1} \pm 32\ \mu g\ m^{-2}\ s^{-1}$). While prior studies indicate the change in bedding may [33] or may not [34] result in a change in $NH_3$ emissions in the first several days, these studies focused on the effect of bedding while in use (and its ability to keep urine and feces separated). Once the bedding is mixed into the waste stream, it is no longer able to separate liquid and solid components of the waste and would likely have no effect on the $NH_3$ emissions of the stored waste as observed.

### 3.2. Seasonal and Annual Emissions

The estimated annual $NH_3$ area-based emission at the WI basin at 30 $\mu g\ m^{-2}\ s^{-1} \pm 15.2\ \mu g\ m^{-2}\ s^{-1}$ was only 11% greater than that at the IN lagoon of 27 $\mu g\ m^{-2}\ s^{-1} \pm 14.6\ \mu g\ m^{-2}\ s^{-1}$ (Table 3). These emissions are comparable to reported cattle slurry emissions [5]. These values are similar to the baseline emissions extracted from a large number of cattle and dairy studies of lagoon emissions (33 g $m^{-2}\ s^{-1}$) but less than that for the noncrusted slurry (53 $\mu g\ m^{-2}\ s^{-1}$) [32]. Representativeness of the measurements in estimating the mean annual daily emissions were assessed by comparing the total unclassified mean daily emissions to the mean seasonally classified daily emissions (Table 3). The total valid daily measurements differed from the seasonally classified mean annual emissions by 17% at the WI dairy and by 3% at the IN dairy: the error in mean annual emissions for the WI dairy is likely much greater than that for the IN dairy.

Daily emissions at both the IN and WI dairies were highest in the summer, lowest in winter, and similar during spring and fall (Table 3). Although the summer mean daily emissions were significantly different (Student' *t*-test, $\alpha = 0.05$) between the 59 $\mu g\ m^{-2}\ s^{-1}$ at the WI basins and 38 $\mu g\ m^{-2}\ s^{-1}$ at the IN lagoon (Table 3), the relatively few daily measurements (seven) at the WI basins suggest that this difference was not based on a representative sample (Table 2). However, the WI emissions during the summer were similar to baseline emissions extracted from a large number of cattle and dairy studies of noncrusted slurry emissions (53 g $m^{-2}\ s^{-1}$) [32]. The mean summer emissions at the IN lagoon (well-represented by the 59 days of measurements) were lower than those measured over a summer at a free-stall dairy lagoon in Alberta, Canada (59 $\mu g\ m^{-2}\ s^{-1}$) [35].

**Table 3.** Seasonal $NH_3$ emissions.

| Farm | Period | NH$_3$ Emissions | | | | |
|------|--------|------|------|------|------|------|
| | | Mean g NH$_3$ s$^{-1}$ | SD [‡] g NH$_3$ s$^{-1}$ | Mean µg NH$_3$ m$^{-2}$s$^{-1}$ | Mean g NH$_3$ hd [†−1]d$^{-1}$ | Mean g NH$_3$ AU [*−1] d$^{-1}$ |
| WI | Spring | 0.20 | 0.105 | 28.8 | 31.6 | 24.9 |
| | Summer | 0.42 | 0.137 | 58.7 | 64.3 | 50.6 |
| | Fall | 0.17 | 0.075 | 24.6 | 24.3 | 21.0 |
| | Winter | 0.07 | 0.015 | 9.2 | 10.1 | 8.0 |
| | Annual | 0.21 | | 30.3 | 40.0 | 32.1 |
| | Total | 0.22 | | 31.2 | 33.1 | 26.8 |
| IN | Spring | 0.24 | 0.033 | 28.4 | 8.96 | 6.79 |
| | Summer | 0.39 | 0.039 | 38.2 | 14.2 | 9.27 |
| | Fall | 0.26 | 0.086 | 24.3 | 9.52 | 6.52 |
| | Winter | 0.17 | 0.016 | 17.8 | 5.22 | 4.47 |
| | Annual | 0.26 | | 27.2 | 10.0 | 7.08 |
| | Total | 0.27 | | 30.4 | 9.47 | 6.76 |

[‡]: SD = standard deviation. [*]: animal unit (1 AU = 500 kg live mass). [†]: hd = animal "head".

Mean winter daily emissions for both the WI basins and the IN lagoon were near or lower than the estimated MDL since the storages were commonly frozen during the winter. All valid measurement days during the winter in WI were on days with a frozen surface. The mean daily emissions on days during the winter when the lagoon was not frozen at the IN lagoon (7 of 52 measurement days) was 13.7 µg m$^{-2}$ s$^{-1}$, essentially at the MDL. Effectively, daily emissions during the winter were negligible regardless of the surface being frozen or not.

Mean daily fall and spring emissions were not significantly different at the two dairies on an area basis but differed in terms of AU or animals (Table 3). Mean daily spring emission from the IN lagoon (Table 3) was much lower than that for the Canadian slurry tank [5]. Mean daily summer emission from the IN lagoon (Table 3) was much lower than that for previously reported WI dairies (51 g hd$^{-1}$ d$^{-1}$) [10]. Presumably, the lower emissions at the IN dairy were related to the manure only coming from the milking parlor and holding area while the other studies included waste from barns.

The mean daily spring emission at the WI basins (Table 3) was lower than that reported for a slurry storage tank in Canada (39 g hd$^{-1}$ d$^{-1}$) [5] although, again, the number of valid measurement days was limited and the storage systems differ greatly (Table 2). Although the sample size was relatively small (Table 2), emissions from the WI storage facility during the summer (Table 3) were similar to previously studied WI dairies [10]. The mean daily fall emission at the WI dairy (Table 3) was also in agreement with that reported for similar WI dairies [10]. While the waste in the basins was from the milking parlor and holding area as with the IN dairy, the waste in these basins was also from the barns, similar to previously reported studies [5,35].

Since air temperatures and wind conditions differed between the two locations (Table 1), an assessment of differences in the emissions of the two manure storages (Table 3) must consider if the difference in air temperatures and wind conditions account for the differences in mean daily $NH_3$ emissions. Variability in seasonal emissions at both waste storage facilities when the surfaces were not frozen was largely a result of temperature and wind speed variations. The coefficient of variation (CV) of the seasonal $Q_{NH3}$(25,2) emissions at the WI storage pits decreased from 55% to 7%. Similarly, the CV at the IN lagoon seasonal $Q_{NH3}$(25,2) emissions decreased from 39% to 9%. Spring had the highest seasonal $Q_{NH3}$(25,2) emissions suggesting that an accumulation of undecomposed manure and $NH_4^+$ concentrations during the winter might have increased the biological activity in the spring as liquid temperatures increased and the surface was no longer frozen [36,37].

The mean annual $Q_{NH3}$(25,2) emissions for the nonfrozen WI basins (80.1 µg m$^{-2}$ s$^{-1}$) was 37% greater than that for the nonfrozen IN lagoon (58.5 µg m$^{-2}$ s$^{-1}$). The greater emissions on an area basis for the WI dairy manure storage was likely in part due to greater accumulation of solids at the WI basins relative to the IN lagoon [12] and greater $NH_4^+$ in those solids [30]. Solids were stored for on average 182 days at the WI dairy storage basin and 31 days at the IN dairy settling pit between removals. There was greater crusting of the WI basins over the IN lagoon (Table S1), which is indicative

of greater solids content [5]. Greater solids typically result in crusting, as reported above (Table S1). The $Q_{NH3}(25,2)$ emissions from the WI basins were not however significantly influenced by the crusting of the surface (unpaired t of 0.29, critical t for two-tailed $\alpha = 0.05$ of 2.05).

The estimated annual emission from the IN lagoon was much less than the WI dairy emissions (Table 3). The cumulative estimated annual $NH_3$ emission when the surface was not frozen was 3.3 kg $hd^{-1}$ $y^{-1}$ (2.4 kg $AU^{-1}$ $y^{-1}$) $^{-1}$) at the IN lagoon compared to 15 kg $hd^{-1}$ $y^{-1}$ (12 kg $AU^{-1}$ $y^{-1}$) at the WI basins. Although there were relatively few daily emission measurements for the summer at the WI basins (Table 2), the similarity of the summer mean daily emissions and those from other studies (discussed previously) suggests that the summer mean daily emission is representative of summer conditions. A comparison of $Q_{NH3}(25,2)$ emissions at the WI and IN storage facilities limited the influence of differences in the temperature and wind conditions: the annual accumulated $Q_{NH3}(25,2)$ emissions were 5.5 kg $AU^{-1}$ $y^{-1}$ for the IN lagoon and 25 kg $AU^{-1}$ $y^{-1}$ for the WI basins. Clearly, differences in the temperature and wind environment did not account for the emission differences.

Design capacities and dimensions of the storage facilities can influence the emissions of stored manure. Assuming equal N excretion rates per AU at the two dairies, emissions could be influenced by the storage volume per unit loading. The ratio of storage volume per AU loading was also larger for the WI basins (24 $m^3$ $AU^{-1}$) than the IN lagoon (13.7 $m^3$ $AU^{-1}$). This would suggest that the WI basin manure solution should be more dilute than the IN lagoon, which was not supported by the visual crusting or the lack of significance in emissions between days with separation and days without separation (Table S1). Consequently, the separation of liquid and solids at the WI dairy was likely less effective than that at the IN dairy. Considering that only liquid manure (urine and flush water) is transferred to the IN lagoon from the milking parlor and holding area and that urine typically represents 40% to 50% of excreted N [4,38], the emissions per total manure loading at the IN lagoon was approximately 50% less than would be expected if both liquid and solids were loading the lagoon. At the WI dairy, the excreted manure from the milking parlor, holding area, and barns was separated most of the time; however, it appeared that a portion of the solids was transferred to the storage basins based on the observed crusting (Table S1). Organic N, expected to be greater in the WI basins than the IN lagoon as a result of the greater DM content, would have contributed to $NH_3$ emissions due to N mineralization [39]. Thus, this difference in volumetric loading (greater for the WI basins than the IN lagoon) and differences in manure characteristics (higher DM and Organic N in the WI basin relative to the IN lagoon) likely accounted for some of the emission differences between the storages.

Annual $N-NH_3$ emissions at the IN lagoon corresponded to 10% (1% per 1% DMI) of the estimated feed N intake of the herd. The low percentage of feed N intake in the IN lagoon emissions was partly due to the lagoon holding only liquid waste, which is typically 40% to 50% of excreted N [4,36]. Even assuming only 50% of the excreted N loaded the lagoon, the estimated total manure emissions relative to intake N would be about 20%, still lower than a compilation of studies (35% to 44%) reported by Liu et al. [40] and much lower than the 72% to 76% reported for dairies in the mid-Atlantic region of the US [41,42]. The reason there was a relatively low percentage of N intake in the lagoon emissions is unknown. Although untestable with the existing measurements, there may have been less excretion while cows were in the holding area and milking parlor than in the barns where they rest and feed [43,44].

Annual $N-NH_3$ emissions at the WI basins corresponded to 40% (2% per 1% DMI) of the estimated feed N intake of the herd. This intake percentage might be expected to be higher than that of the IN dairy since the manure loading the basins are from the milking parlor, holding area, and the barns. This percentage of feed N excreted and emitted by the basins at the WI dairy was within the range of compiled values [40] but much lower than that of dairies in the mid-Atlantic region of the US [41,42]. Partial separation of the liquid and solid manure loading the basins may explain a portion of the differences between dairies.

## 4. Conclusions

Area-normalized $NH_3$ emissions were higher at the WI dairy than at the IN dairy. This difference in emissions was likely partially due to a greater source strength in the WI basins compared to the IN lagoon. Results did not support other studies that indicate that the separation of solids from the manure prior to liquid storage reduces $NH_3$ emissions from the manure storage when referenced to the loading. The emissions were strongly influenced by the temperature at both the IN lagoon and the WI storage basins. Emissions from the IN lagoon were weakly (WI) or not (IN) correlated with wind speed, suggesting only a small influence of emissions on atmospheric mixing. The effective surface area for emissions per manure loading of the WI storage basins was greater than that of the IN lagoon and may contribute to the greater emissions from the WI basins.

**Supplementary Materials:** The following are available online at http://www.mdpi.com/2073-4433/11/10/1108/s1, Table S1: Visual manure storage characteristics, Table S2: Comparison of emission methods (modified from [17]), Table S3: Producer activities at the IN dairy, Table S4: Producer activities at the WI dairy.

**Author Contributions:** Conceptualization, R.H.G. and M.T.B.; methodology, R.H.G.; software, M.T.B.; validation, R.H.G. and M.T.B.; formal analysis, R.H.G. and M.T.B.; investigation, R.H.G. and M.T.B.; resources, R.H.G.; data curation, R.H.G. and M.T.B.; writing—Original draft preparation, R.H.G.; writing—Review and editing, M.T.B.; visualization, R.H.G.; supervision, R.H.G.; project administration, R.H.G.; funding acquisition, R.H.G. All authors have read and agreed to the published version of the manuscript.

**Funding:** This research was funded by livestock producers, the Agricultural Air Research Council, Inc. and Dairy Research Incorporated.

**Acknowledgments:** Thanks go to the extensive field help from Jenafer Wolf, Alfred Lawrence, Scott Cortus, Benjamin Evans, Chris Fullerton, Derrick Snyder, Hans Schmitz, and Gianna Hartman.

**Conflicts of Interest:** The authors declare no conflict of interest. The funders had no role in the design of the study; in the collection, analyses, or interpretation of data; in the writing of the manuscript, or in the decision to publish the results.

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
