# Peer review of "Ammonia Emissions from Differing Manure Storage Facilities at Two Midwestern Free-Stall Dairies"

_atmosphere, doi:10.3390/atmos11101108_

Round 1
Reviewer 1 Report
Although the authors answered most of the questions on the first version of the manuscript. I still have some comments:
- The terminology “The animal-normalized mean annual daily” needs to be clarified in the abstract
- It is said that “ It is hypothesized that although the timing of emissions varies between anaerobic lagoon and slurry storage the emissions will be similar (null hypothesis)”. This hypothesis is not logic because not only the timing of emissions but also manure characteristics and microbial dynamics may differ between anaerobic lagoon and slurry storage.
- There is still a need to explain how visual observations by field personnel, give different percentages of crust on manure surface?. If you have these pictures could you use image analysis to determine the percentages of scum?. In fact, it is hard to see the difference between picture a and picture B in Figure 1.
- It is stated that “The IN lagoon emission estimates were nearly evenly distributed throughout the year”. How could you determine this? Why was not this pattern true for the WI site?
- It is mentioned that “the exponential diffusion function of (Equation 1) accounted for 58% (adjusted R2=0.58) of the variability….”. this sentence should be revised to indicate that the temperature was the dependent factor and not the equation, right?
- What does “Consider first the potential differences in concentration gradient over the manure storage surface” mean?
- “Based on a compilation of studies”, please cite these studies
- “differences in the emissions of the two manure storages (Table 3) must consider if the..” needs to be revised
- "accumulation of un-decomposed manure during the winter might have increased the biological activity in the spring”. How long would it take to establish biological activity after winter time?
- There is a need to explain the differences in the organic and volumetric loadings to manure storages and their effects on the emissions
- “The cause for the low percentage of N intake in the lagoon emissions is unknown”, this sentence is not clear.
- “Although unmeasured, literature suggests that a likely greater NH3 concentration at…”, how can you use this sentence as a conclusion?. Why do you think NH3 concentration would be higher at storage surface?
Author Response
Attached comments to reviewer 1

Reviewer 2 Report
Good response to comments.
Author Response
no comments to respond to
This manuscript is a resubmission of an earlier submission. The following is a list of the peer review reports and author responses from that submission.
Round 1
Reviewer 1 Report
This is a very important topic and a large amount of time, energy and money went into the research. However I have several concerns. Most important is the amount of adjustments in the data and the assumptions. It seems that everything is varying at each location. Was anything consistent?
line 31 I strongly object to the word inefficient. Dairy cows are efficient in converting low value proteins into milk!
line 53 You attempted to make measurements over a year. Did you accomplish that?
lines 70 & 79 This suggests the cows weighed less than 500 kg.Your reference lists cows at 630 kg. I assume they were Holsteins. I am very concerned about your adjustment to AU. You use the information from reference 27 because you seem to have no information about the cows or diets. If the the cows were Holsteins and you estimate the weight at 630 kg then each cow is 1.26 AU. Unless I am missing something, your calculation is incorrect.
line 227 36(2) over nearly 2 years? That does not seem like a good representation. Most in spring ans fall? Good sample?
line 355 can you make this statistical comparison? I assume you are using day as the experimental unit. Actually it is location. Also you are reporting summer emissions but at line 227 you state most at WI were spring and fall.
lines 389-397 Time adjusted is a big assumption. All of this assumes a lot.
lines 398-400 110% ? How can this be taken as reasonable?
line 403 This is reasonable but few measurements.
lines 409-410 My point?
line 412 What is "area normalized" ? Need to emphasize that this is numerical not statistical.
lines 412-417 But everything is varying, not just solids separation.
lines 422-425 Adjustment for time? Assumes a lot.
line 425 Numerically higher and you are comparing 170(144) measurements to 36(2).
Table 2 I have trouble accepting the difference between WI and IN. I calculate a 9 to 11% loss of N from manure in WI and 39-49% in IN depending upon which reference I use for manure production. People would reference these numbers and I am not confident in them.
Reviewer 2 Report
General comments
The research reported in this manuscript is about monitoring ammonia emissions from manure storages. This research is important and timely for determining the environmental impact of dairy farms. However, the research needs and objectives are briefly explained in the introduction. More details about the research gap is needed. There is a need to explain the difference between storage basins, settling pits, and lagoons. Usually, settling basins are used to remove a significant portion of manure solids before lagoons that are the main manure storages in dairies.
It is not clear how did you assume the error for the bLS error to be 20%?. The data was collected in 2007-2008, do you think any changes in dairy management or feed strategies that might happen during the last 12 years could affect the emissions of NH3?. How would the average ambient temperature and wind speed over the last 12 year be differ from the average values reported during the measurements period reported in this manuscript?. How could you determine the percentage of crust or scum over manure storages?. How about the thickness of the scum/crust?. It is not clear why there were emissions during freezing times?. Can not frozen manure surface form a shield against emissions?. You assumed that the minimum detection limit of bLS to be 16.7 g m-2s-1 that was greater than the average value (15.5 g m-2s-1 ) reported for WI dairy. How could this be possible?. There is a need to explain, why would the depth of manure storage affect the temperature if the temperature was measured at 0.3 m depth. Yes, I agree that there should be a stratification of temperature in manure storages. But no data was presented. Air temperature may be the main factor affecting the temperature of storage at 0.3 m depth. No data was presented for the depth of manure storage. No data was also presented for ammonium concentrations in manure storage to support the discussion about the gradient of ammonium concentrations and the effects of pH and ammonium concentrations on the ammonia emissions from the studied sites. Considering the accuracy of the bLS or VRPM emissions model, a clarification is needed about the effect of pH and concentration of ammonium on ammonia emissions. The effect of temperature and wind speed on the emissions were independently discussed. A multiple regression for predicting the emissions as a function of both temperature and wind speed may be more relevant. It is stated that “There was no significant difference in the emissions between the day before, day of, and day after cleaning events”. However, no statistical analysis was performed to support that statement. The SD of the emission was at least 27% (6 g m-2 s-1/22 g m-2 s-1) of the average values. Large values of SD might indicate that there was significant differences. A statistical analysis is needed.
It is mentioned that “high solids content in the WI basins was also evident from the greater crusting of the WI basins over the IN lagoon (Table S4)”. It is not clear how the crust percentages were determined. There is a need to distinguish between percentages of area covered by crust/scum and the thickness of the crust/scum and their effects on the emissions. It is also stated that “The overestimate of N emissions relative to feed input was likely a combined result of higher NH4+-N in milking center manure than manure from other parts of the farm and a larger fraction of manure produced in the milking center relative to other parts of the dairy”. Again, no data is presented for manure characteristics to support this statement. The argument about the greater emissions from IN lagoon than the WI dairy basins is not realistic because the area is not the only factor affecting the emissions from both sources. Other factors can affect the emissions such as temperature, microbial activity, wind speed, manure characteristics. In the conclusions, it is stated that “greater NH3 concentration at the surface of the WI basin compared to the IN lagoon”. No data is given on ammonium concentrations in the surface of the studied manure storages. The conclusion about the effect of the solid separation on the emission is not valid because you mentioned that the separator often failed and you have not given details about the performance of the manure separator.
Minor comments:
- Lines 31-33, for the N losses, please specify which livestock type and what kind of feed.
- Line 37: “ Manure and flushed bedding” may be revised to “flushed manure and bedding”
- Line 39, “well mixed” how?. Do you think that natural mixing can achieve well mixing?
- Line 50, hd=animal head as you defined it on line 388.
- What was the type of liner in WI manure storage?
- Line 69, how did you calculate time-weighted population?
- Line 101, 2.5K2, what is the superscript 2?
- Line 123, “It was assumed that if the surface was frozen….” Needs to be revised. It is a confusing sentence.
- Line 129, please define VRPM
- Lines 133-134, please list the indicators you applied
- Lines 161-162, how did you calculate β (the temperature influence on the emission)
- Line 181-183, how could you validate the measurements for the 50% of the hourly measurements within a 24 h day.
- Lines 187 and 293, change NH3 to NH3
- Line 190-191 and Table S4, no color nor the depth is mentioned in the Table. How could you determine the percentage of crust (e.g., 5%, 80%-85% crust)?
- Line 192, there is a need to clarify that what you determined for the N losses was in the form of NH3 not other N- containing gases
- Line 227 and 228, what are (2) and (144)?
- Line 235, how close was the storage from the barn?
- Please change the data on Figure 1 and Figure 2b to be μg m-3.
- Line 247, “in daily”, “in” can be deleted
- Lines 261 and 262, you may delete “units”
- Lines 260-262, what do 63% and 53% mean? Are they determination coefficient?
- Was equation 1 used by Grant et al. [25]?. What kind of manure storage was employed in the hog operation that was studied by Grant et al.[25]?
- Line 394, no Table 4 is presented
- -Table S1, what is separator pit?